# What Confucian Eco-Ethics Can Teach Us about Solving the Dilemma of Interpreting the Concept of Sustainability

Xian Li and Fuming Wei *

Department of Philosophy and Science, School of Humanities, Southeast University, Nanjing 210096, China; 230179674@seu.edu.cn
* Correspondence: 101001804@seu.edu.cn

**Abstract:** Sustainability is at the heart of the concept of the common home. By prioritizing sustainability, we can create a better common home and ensure the well-being of present and future generations. However, there is a dilemma in the interpretation of sustainability, which is mainly characterized by the irreconcilability between "weak sustainability" and "strong sustainability". The dilemma is partly rooted in some Western philosophical traditions such as the Western separatist mindset, anthropocentrism, and technological solutionism, which have contributed to human subjugation. This paper proposes Confucian eco-ethics to resolve this dilemma. First, Confucian eco-ethics embraces the holistic worldview of "anthropocosmic" that establishes an ontological understanding of the interconnectedness and interdependence between humans and nature, which transcends the Western dichotomy of subject and object and resolves the dualism between human beings and nature. Second, Confucian eco-ethics advocates "pushing oneself to all things" and considers human beings and nature as an ethical community, which emphasizes the ethical responsibility of human beings to protect nature, thus remedying the dilemma that anthropocentrism and ecocentrism have too little or too much responsibility for nature. Third, Confucianism endorses benevolence as a core value for managing technology to achieve sustainable development, and it favors a comprehensive approach that combines technological innovation, values reform, and institutional reform to solve ecological problems. To do this, we analyze the Dujiangyan Water Hydro-Project Hydraulic Project as a case study to illustrate the practical feasibility of Confucian eco-ethics in achieving sustainable development. The conclusion suggests that Confucian eco-ethics can enrich and expand sustainability theory, offering an alternative pathway for a better common home.

**Keywords:** sustainability; common home; Confucian eco-ethics; "anthropocosmic"(天人合一); the Dujiangyan Water Hydro-Project

## 1. Introduction and Research Questions

Sustainability is at the heart of the concept of the common home (Colglazier 2015). By realizing the goals of sustainable development, we can ensure that the Earth becomes a lasting and beautiful common home, providing a sustainable environment and society for current and future generations (Francis 2019, pp. 503–10). Sustainability has emerged as a fundamental core value in modern society, exerting a profound and enduring impact on individuals, companies, and nations (Kuhlman and Farrington 2010; Horlings 2015). Achieving harmonious development of the economy, society, and environment requires a proper understanding of sustainability and the implementation of appropriate actions. By doing so, we can effectively work towards creating a future that is both sustainable and prosperous for the world.

Since the Brundtland Commission first formally introduced the concept of sustainability in 1987, the controversy over its interpretation has not ceased. The Commission considered sustainability to be related to sustainable development, defined as "development that meets the needs of the present without compromising the ability of future generations

to meet their own needs" (WCED 1987). However, this definition has been criticized for its narrow focus on human needs and its limited consideration of the ethical relationship between human beings and the natural environment, as well as the sustainability of other organisms (Shiva 2005; Johnston et al. 2007).

These criticisms became the starting point for contemporary controversies over the interpretation of the concept of sustainability, particularly in the Western context. One notable contribution to this discourse is the distinction made by Beckerman (1995) between weak and strong concepts of sustainability. The "weak sustainability" emphasizes the use of resources in a manner that allows for their regeneration and replacement, while the "strong sustainability" goes further to emphasize the intrinsic value of nature and the need to protect it for its own sake and not just for human benefit.

The controversies surrounding sustainability can be traced back to several key issues. (1) Which has a higher priority: human or nature? (2) Whether humans have a moral responsibility and obligation to nature. (3) The extent to which the use of technology can solve ecological problems, etc.

These controversies partly stem from some Western philosophical traditions, including subject–object separatist thinking, anthropocentrism, and technological solutionism, which have arguably contributed to the plundering of nature by mankind and triggered the global ecological crisis. In order to truly liberate ourselves from this global crisis, humans must draw on multiple resources with a more open attitude and make up for the shortcomings of Western sustainability theories so as to build a sustainability theory adapted to modern society.

Therefore, an alternative path to enrich and improve the theory of sustainability is by complementing the deficiencies of the Western paradigm of sustainability concept from the perspective of Eastern Confucianism (Mak and Cheung 2014; Guo et al. 2017; Crippen 2023; Wang et al. 2023; Wong 2023). In addition, Crippen (2021) has enriched this theory from the perspective of African philosophical traditions. Confucianism is rich in ecological ethics and adheres to the holistic mode of thinking of "the unity of heaven (nature) and human", considering humans and nature as an organic whole, and extending ethical responsibility to nature through the "extend oneself to others" approach, thus regarding humans and nature as an ethical community. Furthermore, Confucianism advocates virtuous governance of science and technology with benevolence and righteousness as the core values of ecological ethics and the use of ritual and law as institutional support to suppress the various negative values generated by the development of science and technology.

The primary objective of this paper is to investigate the conflict within the Western explanatory paradigm of the sustainability concept and propose a resolution using Confucian eco-ethics to reconstruct the concept of sustainability. The research framework comprises the following key components: an analysis of the conflict within the Western explanatory paradigm of sustainability, including an examination of its underlying causes (Section 2); exploration of Confucian eco-ethics and its significant contributions to sustainability theory (Section 3); illustration of case studies that exemplify the application of Confucian eco-ethics in achieving sustainable development (Section 4); and the last section provides conclusions and an overview of possible future work (Section 5).

## 2. Theoretical Controversies of Sustainability and Their Causes

The historical development of the interpretation of the concept of sustainability in the West has been characterized by a growing controversy between "weak sustainability" and "strong sustainability".

The concept of "weak sustainability" was developed by Pearce and Atkinson (1993), supported by Robert Solow (2014), and subsequently enriched by Gutés (1996). "Weak sustainability" refers to the use of man-made or other forms of capital to replace natural resources while meeting current human needs and economic growth in order to maintain overall welfare and economic growth. Weak sustainability represents an anthropocentric view of sustainability.

In contrast, "strong sustainability" emphasizes the long-term sustainable development of humans and nature based on maintaining ecological integrity. This view holds that natural capital (referring to the earth's natural resources and ecosystems, including air, water, forests, biodiversity, and other elements that provide essential services that support life) has intrinsic value and function and that humans do not have an inherent right to exploit nature (Devall 1990). The preservation of the integrity, stability, and beauty of natural ecosystems should be the ultimate goal and measure of human moral behavior (Rolston 1988; Taylor 2011).

## 2.1. Theoretical Controversies

Although both "weak sustainability" and "strong sustainability" agree with the Brundtland report's overall goals for human development and environmental protection (WCED 1987), there is still controversy between the two regarding the priority of humans and nature: whether humans have a moral responsibility and obligation to nature and the extent to which the use of technology can solve ecological problems. The details are as follows.

The controversy between "weak sustainability" and "strong sustainability" is centered on who has more priority, humans or nature. "Weak sustainability" holds an anthropocentric position, emphasizing the "human as the subject and nature as the object" model, which places human needs and interests above the protection of the environment and ecosystems. In contrast, the "strong sustainability" approach takes a nature-centered stance, emphasizing the "nature as the subject and human as the object" model, prioritizing the environment and placing the values and rights of nature above the economic and social interests of humans.

Whether nature has intrinsic value, or whether humans have a moral responsibility and obligation to nature, is also a point of conflict between "weak sustainability" and "strong sustainability". The question of intrinsic value is an important ethical principle in protecting individual organisms from being destroyed (O'Neill 1992; Jamieson 2002). Scholars with a weak sustainability perspective believe that only anthropocentric Homo sapiens species have intrinsic value, while non-perceptual objects in nature, such as plant species, rivers, mountains, and landscapes, have no intrinsic value and can only be seen as instrumental values that serve human needs. In contrast, scholars with a strong sustainability perspective argue that both individual creatures (including animals, plants, and microorganisms) and natural systems as a whole, which exist independently of human life centers, have an intrinsic value equal to that of humans and should therefore be respected (Rolston 1988; Agar 2001; Brennan and Lo 2010; Taylor 2011).

Based on the premise that nature has no intrinsic value, "weak sustainability" holds that humans have no natural moral responsibility or obligation to nature. The ultimate goal and measure of human moral behavior is to ensure the long-term survival of humanity as a whole, viewing nature as a resource for achieving human goals (Devall 1990). In contrast, strong sustainability holds that nature has intrinsic value and that humans do not have an inherent right to exploit nature. Therefore, humans should be morally concerned with natural processes and ecosystems. Their moral responsibilities and obligations towards nature should not be based on satisfying human interests or needs but rather on the interests of nature itself and the intrinsic value of natural things.

The extent to which the use of technology can solve ecological problems is also a point of conflict between weak and strong sustainability. Weak sustainability relies excessively on technological innovation to solve ecological problems while maintaining existing production and consumption patterns. However, the application of technology only temporarily alleviates some ecological problems without fundamentally changing the overuse of resources and environmental damage.

*2.2. Causes Analysis*

The current interpretation of the concept of sustainability is dominated by the Western paradigm, and there is a controversy between "weak sustainability" and "strong sustainability". This controversy may lead to the emergence of extreme views and thus irreconcilable dilemmas. At the root of the controversy are the subject–object dichotomy, anthropocentrism, and the limitations of technological solutionism, all of which are inextricably linked.

The Western subject–object dichotomy has resulted in a lack of integrated systems thinking when it comes to understanding the relationship between humans and nature. Within the subject–object mindset, which is prevalent in the West, there is a tendency to separate and oppose humans and nature, subject and object. Both weak and strong sustainability approaches lack integrated thinking about the entire system of humans and nature, resulting in a preference for one side over the other.

The anthropocentric bias has resulted in the trivialization of environmental issues, disregarding the complexity of ecosystems and the significance of biodiversity. In current sustainable development practices, the West continues to prioritize human economic development and well-being, emphasizing economic growth and placing the protection of the environment and natural resources on the back burner. Economic growth is often viewed as the primary goal of sustainable development, while the impact of economic growth on resource consumption and environmental loads is ignored. This prioritization has resulted in inadequate attention being paid to environmental issues.

The bias towards technological solutionism has resulted in the belief that technology plays a decisive role in driving social and economic development. This approach looks to technological innovations and solutions to solve environmental problems while ignoring changes at social, cultural, and institutional levels. Technological determinism ignores the impact of social and political factors on sustainable development. The choice and application of technology are often influenced by factors such as power, economic interests, and social preferences and do not depend entirely on the characteristics of the technology itself. Moreover, relying solely on technological advances may not solve fundamental environmental and sustainability challenges and may even lead to more serious ecological problems.

These factors influence the perception, interpretation, and choice of solutions for sustainability. The key to resolving the controversy between "weak sustainability" and "strong sustainability" lies in finding a balance between humans and nature. This means integrating human needs and interests while protecting the health of the natural environment and ecosystems. Therefore, finding solutions and resources from Eastern Confucianism becomes one of the available paths.

## 3. Confucianism's Eco-Ethics and Its Contribution to Sustainability Theory

Confucianism is one of the most important schools of traditional Chinese philosophy. Although sustainability is not directly discussed in Confucian texts, they contain a wealth of eco-ethical ideas that have been extensively explored by scholars (Callicott and Ames 1989; Tu 1998, 2010; Gao 2015; Zhang 2023). Confucianism's eco-ethical thinking is rich, systematic, and innovative, including an organic and holistic view of the universe and nature, the core idea of the unity of heaven and humanity, and the moral relationship between humans and nature. This section focuses on the ontological foundations, extended ethical responsibilities, core values, and institutional support of Confucian eco-ethics. Based on this, it further discusses how Confucian eco-ethical thinking can resolve conflicts in sustainability concepts within Western interpretative paradigms and contribute to the enrichment and development of sustainability theory.

*3.1. Ontological Foundation: "Anthropocosmic" ("The Unity of Heaven and Human")*

"Anthropocosmic" (Tu 2010), which refers to "the unity of heaven and human", is the core concept of Confucianism's ecological ethics. It emphasizes the organic wholeness

and interdependence of humans and nature. The holistic thinking of "the unity of heaven and human" serves as the ontological foundation of Confucian eco-ethical thought. The concept of "Anthropocosmic" encompasses both "heaven" and "human", where "heaven" refers to the natural world and "human" refers to human beings. Grim and Tucker point out that "for Confucians, humans are anthropocosmic beings in relationships, not anthropocentric individuals in isolation" (Grim and Tucker 2014, p. 121).

The Confucian concept of "the unity of heaven and human" regards humans and nature as a unified, organic, interactive whole (Callicott and Ames 1989, pp. 67–78). Confucianism believes that all things in the universe, including the elements of heaven, earth, human beings, and things, are composed of "*qi*" (气). Under the influence of "*qi*" (气), all things in the universe are interconnected and interact with each other, displaying a process of continuous change and development. "Heaven" refers to the source of life and value for human beings and all things, which is a sacred and supreme being. Confucianism maintains that there is an order and a moral code in heaven called "the way of Heaven." The moral code of conduct and behavior of human beings in the universe is called "the way of Human". Confucianism describes the concept of heaven as the expression of the purpose of nature and the source of goodness and benevolence. As Zhong Yong said, "what Heaven mandates is called 'the nature'; to follow the nature is called 'the Way'; to cultivate the Way is called 'instruction'" (Ivanhoe 2019, p. 191). Confucianism emphasizes that human beings should follow the requirements of the divine principle in the universe to achieve personal moral cultivation and the harmonious and sustainable development of society. Confucius asserted the following: "The gentleman has three things he stands in awe of. He stands in awe of the Mandate of Heaven, of persons in high position, and of the words of the sages" (Watson 2007, p. 116). Therefore, people should respect heaven, which is not based on religious belief or superstition, nor is it a fear or compromise of God's authority, but rather a moral cultivation. Confucianism suggests that people recognize and gain insight into the meaning of heaven through moral cultivation and practice.

Furthermore, Confucianism regards humans and nature as interdependent and mutually influential. According to Confucianism, human beings are a part of heaven (nature), but they are also a special part of nature. This is manifested in the fact that humans can exert their subjective initiative and consciously recognize, grasp, and use the laws of nature to manage all things. All of this is done to nourish all things and promote their own development, ultimately leading to the realization of harmony and unity between human beings and nature. On the one hand, heaven (nature) endows human beings with life and human nature. As Xunzi stated, "Heaven and Earth produce the noble person" (De Bary et al. 1999, p. 169). Zhongyong said that "what Heaven mandates is called 'the nature'" (Ivanhoe 2019, p. 191). Once humans are endowed with "human nature" by "Heaven", they become distinct from other animals and acquire a unique subjective initiative in fulfilling their mission consciously. On the other hand, humans are a special existence in heaven (nature), as they are the spiritual essence of all things and possess nobility through morality. The Confucian belief is that heaven and humans are a unified entity based on the two energies of *yin* and *yang*. This unity exhibits a developmental sequence from the lower to the higher level. Xunzi noted that humans have energy, life, knowledge, and righteousness. Therefore, they are the most precious beings in the world. In other words, humans are the highest form of energy development, possessing life, consciousness, and "righteousness" (morality). The value of humans lies in their morality, which is manifested in their moral obligations and responsibilities to nature, as well as in their ability to achieve harmony and unity with nature through moral means.

### 3.2. Expanded Ethical Responsibility: "Pushing Oneself to All Things"

Ecological ethics is a result of human beings' concern for their own survival and their compassion for all beings in the world. Confucianism's eco-ethics extends the object and scope of morality from human beings to all things in the universe. While affirming that human beings have the highest value, it also emphasizes the need to extend benevolent

care to all things. Confucianism advocates that, on the basis of blood kinship, ethical responsibilities between relatives should be extended outward by means of "push". This is manifested in the three levels of "love for blood relatives-love for the people-love all things in nature".

Confucianism sees the self, others, and all things in heaven and earth as one. As the Song philosopher Zhang Zai (1020–1077) put it, "Heaven is my father and Earth is my mother, and even as a small creature, I find an intimate place in their midst. Therefore, I regard that which fills the universe as my body and that which directs the universe as my nature. All people are my brothers and sisters, and all things are my companions" (Chan 1963, p. 497). The term "all people" refers to individuals other than oneself, while "things" includes all aspects of natural existence, including living things (such as plants and animals) and non-living objective entities (such as mountains, rivers, and rocks). The declaration that all are my brothers and sisters emphasize the oneness of the self with others, while "all things are my companions" signifies the oneness between the self and nature. The concept of the oneness of self and others and self and nature implied by humans and things is extended to include all existence in the universe in order to emphasize and strengthen thiolations as much as possible. The Ming philosopher Wang Yangming (1472–1529) linked the love of people and the love of things to form a complete and unified moral concept covering both interpersonal and ecological ethics, expanding the Confucian ethical relationship between people to an ecological ethical relationship between people and all things (Chan 1963, p. 659).

In short, Confucianism incorporates all things in nature into the realm of moral concern by "pushing oneself to all things" and encourages the extension of benevolence to oneself and the natural world. Confucianism believes that human beings and nature co-exist in the universe and that they are one ethical community. Therefore, it emphasizes that people can extend their moral responsibilities and duties to a wider range of objects, including animals, the environment, and the entire ecosystem. This concept is consistent with the goal of modern eco-ethics, which aims to awaken people's moral consciousness, extend it to the community of life and its members, and give moral status to the community of life and its members. Therefore, humans are responsible not only for learning and nurturing themselves but also for managing all living things. People's actions and decisions should respect and maintain the stability and prosperity of the entire ecosystem.

*3.3. Core Values: Benevolence and Righteousness*

Benevolence and righteousness are not only the most unique spiritual marks of Confucianism but also the core values of Confucian eco-ethics. They provide the value basis for people to protect the environment and achieve sustainable development.

Among them, benevolence is regarded as the highest value of Confucian eco-ethics. Confucian benevolence is the characteristic of humans towards goodness in all things in the world; it is the purpose of what makes man human. Confucius regarded "love for others" as one of the manifestations of benevolence, and Mencius regarded "compassion" as one of the manifestations of benevolence. It can be said that humans must realize the goodness of humans through specific emotions such as love and compassion, that is, benevolence is the goal of the realization of a series of emotions sprouting from love and compassion. Confucianism expounds the concept of benevolence, which is the heart of loving people, an emotion realized through compassion, and explains that human beings must be good and "love things in nature" for humans and nature to co-exist in harmony and prosperity. Confucius believed that "benevolence" itself contains the connotation of symbiosis and co-prosperity with oneself, with others, and with all things. Confucius put it as follows: "What you do not want others to do to you, do not do to others" (Watson 2007, p. 80). "Righteousness" is another core value of Confucian eco-ethics, which refers to the conformity of thoughts and behaviors to certain guidelines through internal self-regulation, and is used as a moral code and code of conduct to judge whether people's behavior is appropriate. Confucianism does not provide a definitive conceptual description of "righteousness" but

understands it as "appropriateness"; Zhongyong stated that righteousness is also appropriate. In addition, "righteousness" is also regarded as the way to realize "benevolence". Benevolence, as the heart of loving people, needs to be expressed in the form of righteousness, that is, the implementation of benevolence according to the appropriate situation. In this way, "righteousness" has become the core value of Confucian eco-ethics, which encompasses both interpersonal and ecological morality.

In conclusion, by integrating benevolence and righteousness into eco-ethical practices, people can establish a more positive and responsible symbiotic relationship with nature and contribute to the realization of ecological civilization and sustainable development.

### 3.4. Institutional Support: Rites and Laws

Confucianism does not reject technology but rather treats it with tolerance; Confucius said the following: "A craftsman who wants to do his job well must first sharpen his tools" (Watson 2007, p. 107). Confucianism uses ritual and law as institutional support to regulate human behavior, thus suppressing the negative values generated by the development of science and technology.

Rites originated from the practices of ancient Chinese tribes. The original purpose of religious festival rituals was to provide a behavioral model for people's religious activities, standardizing and ordering them, and maintaining primitive kinship blood relations with a form of rationality. As the rites developed, they gradually evolved into a system of etiquette regulating all aspects of people's clothing, food, housing, and transportation. The scope of their role expanded to provide an institutional basis for the effective functioning of the state and society as a whole, placing the political activities of the state and the social life of the common people on an established pattern of order. For example, in Zuo Zhuan, it is stated that the role of rites is to organize the state, determine the gods of earth and grain, order the people, and benefit the heirs. Additionally, rites serve as a preventive measure against behavior that is contrary to order. Han Book's Jia Yi's biography said the following: "Rites are forbidden before they are to be, and laws are forbidden after they have been". This means that ethical education can prevent things from happening in the first place, while legal sanctions only provide punishment and remedies after the fact.

It can be said that the role of rites in sustainable development is mainly to regulate people's behavior through the formulation of national and social systems to achieve the purpose of preventing problems before they occur so that the overall ecological environment and the direction of human development will move towards the goal of sustainability. The following is recorded in the Analects: "The Master fished with a rod but not with a longline. He shot at birds with a stringed arrow, but not if they were roosting" (Watson 2007, p. 51). This means that Confucius fishes with a fishing rod but not with a net, and he shoots birds with arrows but does not hunt birds that have returned to their nests and roosts. The purpose of doing so is to maintain ecological balance and to prevent the loss of long-term benefits due to momentary gains. After Confucius, this proposition was widely recognized.

The rite is the way to practice the core Confucian eco-ethical values of benevolence and righteousness. Confucius took "benevolence" as the inner basis of "ritual" and "ritual" as the outer expression of "benevolence". "Benevolence" is more practical because of "ritual", and "ritual" has an inner moral basis and vitality because of "benevolence" so that it does not flow into a rigid and empty form. Ritual is a code for all kinds of relationships (ethical, political, and social) and a norm that regulates the relationship between all things and between humans and nature. It defines the responsibilities and duties of individuals in society, requires people to show reverence, respect, and humility in their interactions with nature, and promotes harmony between humans and nature. Confucius stated the following: "What ritual values most is harmony" (Watson 2007, p. 17). Xunzi believed that harmony is the ecological mechanism that supports the creation and development of all things and that attaining harmony with heaven is essential for all living beings to grow and thrive (De Bary et al. 1999, p. 171).

In addition, the law is a complement to ritual in terms of legal and institutional regulation. In addition to rites, Confucianism advocates the restraint and regulation of human behavior through laws and institutions, which include both the formulation and enforcement of regulations for environmental protection, resource use, and ecological balance and the regulation and supervision of human environmental behavior.

Rites and laws play an important role in Confucian eco-ethics and provide the institutional framework and normative guidelines for achieving ecological conservation and sustainable development. By following the guidelines of rituals, people can establish good relationships with nature and achieve a harmonious coexistence between humans and nature. At the same time, through legal and institutional constraints and regulations, people can protect the environment, use resources rationally, and maintain ecological balance. These institutional supports help promote a balanced and sustainable development between people and nature.

*3.5. The Contributions of Confucian Eco-Ethics to Sustainability Theory*

Confucian eco-ethical thinking offers a different perspective and path for the interpretation of the concept of sustainability, which can help us to go beyond some of the shortcomings of the Western paradigm of interpreting the concept of sustainability.

Confucianism's holistic thinking of "the unity of heaven and human" emphasizes the organic wholeness and interdependence of humans and nature, which transcends the Western dichotomy of subject and object and solves the problem of the dichotomy of human and environment in the Western paradigm of sustainability. The holistic thinking of "the unity of heaven and human" provides a systematic and integrated ontological foundation for sustainability theory. Sustainability theory takes into account not only human interests and needs but is also concerned with preserving the health of the natural environment and ecosystems. It emphasizes the close relationship between human beings and the natural environment and advocates an ethical approach to sustainable development. Thus, Confucian eco-ethics provides a comprehensive and powerful theoretical foundation for the study and practice of sustainability issues.

Confucianism regards human beings and nature as an ethical community and extends the ethical relationship between human beings to an ethical relationship between human beings and all things in nature. It requires human beings to assume not only social responsibility but also natural responsibility to respect nature and protect the ecological environment, which makes up for the shortcomings of anthropocentrism, promotes people to follow the laws of nature, respects the balance and diversity of the ecosystem, stimulates individuals to act positively on sustainability issues, and achieves harmonious development of human beings and nature. At the same time, Confucian eco-ethics is also very different from nature-centeredness in strong sustainability. Although Confucianism emphasizes that human beings are responsible for the ecosystem as a whole, Confucianism emphasizes that "love has its differences", which requires people to be responsible in different ways for different objects and different situations.

Confucian eco-ethics, based on approving the use of technology, uses benevolence and righteousness as core values for the virtuous governance of technology and uses ritual and law as institutional support to suppress the various negative values generated by the development of science and technology so that they can develop in the direction of benefiting all of humanity. Confucianism advocates a combination of technological innovation, values reform, and institutional reform to solve ecological problems. It can be said that Confucianism emphasizes both individual inner moral cultivations to shape a good sense of ecological ethics and moral and institutional norms to ensure people's compliance with environmental protection and sustainable development and to promote ecological progress in society.

## 4. Case: The Dujiangyan Water Hydro-Project

In this section, we discuss the case of Dujiangyan (Figure 1), one of the most famous ancient hydraulic engineering projects in China. The Dujiangyan Water Hydro-Project embodies Confucian ecological ethics and modern advanced flood control ideas, making it a sustainable hydraulic ecological engineering project. It is a well-known case of the combination of Confucian ecological ethics theory and engineering practice (Bangben 2008; Kuhlman and Farrington 2010; Liu et al. 2023). The Dujiangyan Water Hydro-Project is located in Dujiangyan City, Chengdu City, Sichuan Province, China. It was built by the Qin State in 256 BC and is one of the few large-scale public hydraulic engineering projects in the world that has been in use for over 2000 years and still plays an important role today in irrigation, flood control, water supply, and ecology. The Dujiangyan Water Hydro-Project not only contributes to the regional ecological civilization of Sichuan Province in China through its ingenious engineering design concepts and systematic management systems but also contains rich wisdom for sustainable development of major engineering projects, making it a typical case for sustainable development research of major engineering projects (Xiao et al. 2023).

Firstly, the design of the Dujiangyan Water Hydro-Project follows a holistic approach of "unity of human and nature", which extends the engineering beyond the environment and integrates the environment into the engineering, achieving a harmonious coexistence between engineering (human) and environment (nature). The Dujiangyan Water Hydro-Project consists of two important components, namely, the inlet and the canal system. The inlet includes three major structures, namely, the Fish Mouth Weir (diversion), the Fly Sand Weir (flood discharge and sediment removal), and the Bottle Neck (water intake), while the canal system includes a network of artificial canal systems spanning the three major river basins of Minjiang, Tuojiang, and Fujiang. Both the inlet and canal systems of Dujiangyan adopt a no-dam water diversion structure (Feng 2014), achieving a harmonious coexistence between artificial canal systems and natural river channels. In addition, effective coordination between various structures makes the Dujiangyan Water Hydro-Project an organic whole. This not only reduces damage to the natural ecosystem but also manages climate and natural disasters such as floods and droughts. It also solves problems such as irrigation, flood control, navigation, water supply, and ecology, achieving a perfect integration of engineering measures, the natural environment, and human interests.

Furthermore, the Dujiangyan Water Hydro-Project carries multiple objectives such as navigation, irrigation, water supply, flood discharge, and other functions. These functions are accompanied by seasonal changes and social transformations, which are the results of human intervention and regulation. It can be said that the achievement of the multi-objective functions of Dujiangyan relies on human intervention and regulation. When the intervention and regulation are effective, the ecological balance of the Dujiangyan system is maintained, and the regional economy, society, and culture can achieve sustainable development. On the contrary, excessive human intervention can lead to an imbalance in the ecological system of the Dujiangyan irrigation area. For example, from 1878 to 1949, due to political turmoil, frequent floods, inadequate government management, and difficulties in implementing the maintenance system, the inlet structures were damaged, and the irrigation area suffered from frequent disasters, causing significant damage to farmers (Xiao et al. 2023). However, in most periods, due to rational and effective human intervention and regulation, Dujiangyan has continuously played a role in promoting the agricultural economy and commercial development in the irrigation area through irrigation and navigation functions. It also serves as a source of urban water supply and recreational activities to maintain social stability and development. Additionally, it fulfills functions such as flood discharge and ecological water supply to ensure environmental recycling and regeneration. Dujiangyan not only benefited its contemporaries but also left a lasting impact on future generations, demonstrating its strong capacity for sustainable development.

Once again, the successful operation of the Dujiangyan Water Hydro-Project for over two thousand years can be attributed to a comprehensive and continuously improving

"Annual Repair System" that embodies the spirit of Confucianism. People have adapted to the flood and drought patterns of the Minjiang River at different times and varying levels of engineering technology by carrying out routine maintenance, major repairs, special repairs, and emergency repairs. The "Annual Repair System" refers to the annual maintenance carried out during the winter and spring agricultural off-seasons and when water levels are low and easy to operate. This traditional system is followed by all members of society, including officials and civilians, and includes not only routine maintenance but also major repairs every few years or special and emergency repairs as needed. The officials responsible for the "Annual Repair System" are called "Yan Guan"; their tasks include organizing and implementing maintenance and daily operations management of the Dujiangyan Water Hydro-Project. In terms of specific "Annual Repair" implementation, maintenance of the Dujiangyan Water Hydro-Project has been a collective action coordinated by officials and civilians (Feiock et al. 2009). On the one hand, according to historical records, officials directly participated in "Annual Repair" throughout history, including the Qin, Han, Tang, and Song dynasties. On the other hand, people in Sichuan also participated in "Annual Repair" under the official organization, which injected vitality into the sustainable development of Dujiangyan. In the new era, the Chinese government adheres to the concept of green development and has launched a new journey of sustainable development for the Dujiangyan Water Hydro-Project by innovating water conservancy engineering and irrigation management based on past excellent management experience (Xiao et al. 2023).

In summary, the sustainable operation and development of the Dujiangyan Water Hydro-Project rely not only on the holistic thinking of "harmony between nature and humans" in engineering design but also on the management systems and departments established based on the spirit of ritual and law and combined with advanced technology. Furthermore, it depends on the coordinated maintenance activities between the authorities and the public. All of these ensure that the management and operation of the Dujiangyan Water Hydro-Project follow a clear set of rules and regulations, enabling its continuous development and maintenance for the benefit of future generations. The case of Dujiangyan demonstrates the feasibility of Confucian ecological ethics for sustainable development. It integrates holistic thinking with technology and systems and is dedicated to achieving harmonious unity and benign development between humans and nature. This case provides valuable experience and inspiration for us to learn from and apply in today's sustainable development practice.

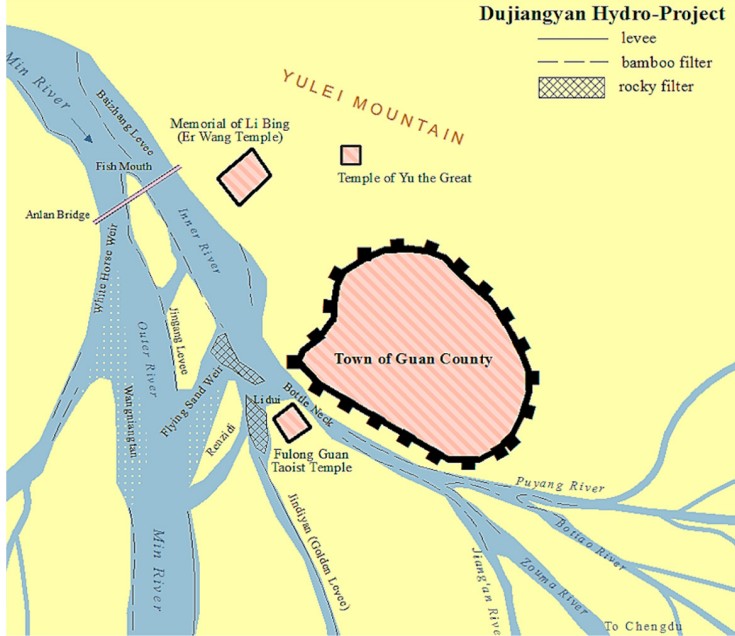

**Figure 1.** Map showing the plan of the Dujiangyan Hydro-Project (Wikiwand, Dujiangyan n.d.).

## 5. Conclusions

In conclusion, it is feasible and fruitful to use Confucian eco-ethics to develop sustainability theory beyond the limitations of the Western paradigm. Through the analysis of Confucian eco-ethics, we find the following:

First, Confucian holistic thinking of "the unity of heaven and human" transcends the Western dichotomy of subject and object and solves the problem of the dichotomy between humans and the environment in the Western paradigm of sustainability.

Second, Confucianism views nature as its own ethical community, which promotes people to follow the laws of nature, respect the balance and diversity of ecosystems, and inspire individuals to take positive action on sustainability issues to achieve harmony between humans and nature.

Third, Confucianism approves of the use of technology based on benevolence and righteousness as core values for the virtuous governance of technology. Confucianism also uses ritual and law as institutional support to suppress the negative values generated by the development of science and technology, and it advocates a combination of technological innovation, values reform, and institutional reform to solve ecological problems, which helps to compensate for the limitations of Western technological determinism.

However, this paper's excavation of Confucian eco-ethical thought is only a beginning, and further research is needed in the future. It is worth noting that any cultural and philosophical tradition has its own unique perspectives and limitations. Therefore, there is also a need to integrate Confucian eco-ethical thinking with modern scientific, economic, and social knowledge and practices in order to establish a more comprehensive and adaptable theoretical framework of sustainability to the needs of modern society.

**Author Contributions:** Conceptualization, X.L. and F.W.; methodology, X.L.; software, X.L.; validation, X.L. and F.W.; formal analysis, X.L.; investigation, F.W.; resources, X.L.; writing—original draft preparation, X.L. and F.W.; writing—review and editing, X.L.; project administration, X.L.; supervision, F.W. All authors have read and agreed to the published version of the manuscript.

**Funding:** This research was funded by the Postgraduate Research Innovation Program of Jiangsu Province [grant number KYCX18_0210] and National Social Science Foundation of China [grant number 21BZX070].

**Acknowledgments:** We would like to express our gratitude to Haoran Jia, for his encouragement and assistance in enhancing this article. We would also like to thank the constructive feedback and insightful suggestions provided by the academic editor and two anonymous reviewers. Their input has significantly contributed to the refinement of this work. We would also like to thank to the editorial team at the Religions Editorial Office for their diligent and meticulous efforts in readying the manuscript for publication.

**Conflicts of Interest:** The authors declare no conflict of interest.

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
