# Peer review of "What Confucian Eco-Ethics Can Teach Us about Solving the Dilemma of Interpreting the Concept of Sustainability"

_religions, doi:10.3390/rel14091216_

Round 1
Reviewer 1 Report
This is a nice paper with interesting and innovative subject matter, i.e., the application of Confucian outlooks to sustainability efforts, as they have played out in China historically and recently. It’s also good that the authors stress things that China has done and is doing right (in light of overwhelmingly negative Western media coverage about environmental and economic matters in China that is frankly propagandistic). Also, in passing, I visited the Dujiangyan area less than a week ago, so enjoyed the paper for this reason too.
However, though the changes I suggest below can easily be handled and likely executed in just a few days, I cannot green light acceptance with minor revisions because the changes are important and therefore (in my view) required. (I would prefer an intermediary category between minor and major revisions, but given this is lacking, I’m compelled to go with major revisions, but note again, I don’t think the authors need to do that much work).
One issue is that the authors fall into a now common trend in recent comparative work on philosophy and sustainability. This is to treat Western outlooks as one monolithic thing, and cast this monolithic thing as the villain. Given the complexity of the history of Western thought, this move immediately disqualifies the article for those who have a good grasp on the history of philosophy, and are aware of both ancient and recent convergences, for example, between Western and East Asian outlooks. In the abstract, the authors are accordingly advised to re-phrase to stress that the dilemma they speak of is rooted in some—but not all—Western philosophical traditions.
The same overstatement is later echoed (56-61) when the authors again affirm that “These differences in perspectives are deeply rooted in Western philosophical traditions, including subject-object separatist thinking, anthropocentrism and technological solutionism, which has led to man’s conquest and plunder of nature and triggered the global ecological crisis. In order to be truly liberated from this global crisis, human beings must draw on multiple resources with a more open attitude and make up for the shortcomings of Western sustainability theories, so as to build a sustainability theory adapted to modern society.” Again, not all Western philosophy stresses subject-object dichotomy, etc. (and several movements do just the opposite). There are also examples of holism all over the place in the West. The authors do not need to rehearse Western philosophy. They just need to soften their claims. It’s ok to say something like “problems are partly rooted in Western traditions.” Leaving in the overstatements will merely serve to disqualify what seems to be a decent paper so far.
Another point: if sustainability problems are overwhelmingly rooted in Western thinking, how do we explain, for example, human-caused desertification in very ancient parts of North Africa and West Asia, or again the fact that humans seem to have driven some species to extinction in prehistoric times? Given that Western philosophy did not exist in prehistorical times, this put stress on the authors’ strong statements.
It's worth adding that by setting up an East-West dichotomy, the authors are in some ways committing some of the fallacies they are criticizing.
Another problem is needless use of gendered language.
For example:
(67-68) “considering man and nature as an organic whole” should be changed to “people and nature” or “human agents and nature” or something not specifically masculine.
Same for:
“has led to man’s conquest and plunder (8-9).
“the relationship between man and nature” (152).
“…separating and opposing man and nature, subject and object. Both weak and strong sustainability lack integrated thinking about the whole system of man and nature, leading to a preference for one side over the other.” (153-155)
“…lies in finding a balance between man and nature. This means integrating man needs… “ (176-177)
“…core idea of the unity of heaven and man, and the moral relationship between man and nature.” (186-187).
“dependence of man and nature” (195)
Bunch more instances between 201-234.
Bunch more between pages 7-9.
Could use more clarification on what it means to follow the “laws of nature” (358)
The distinction between strong and weak sustainability is nicely introduced and explained. Maybe outside scope of paper (or maybe where the paper is going?), but there is a third possibility somewhere between weak and strong sustainability where values are co-constituted through interactions between agents and things like mountains. The idea here would be that values are in ecological arrangements in the world and thus not projected, yet not in any individual thing by itself. Such a view generally works within in East Asian outlook. If this is where the authors are going (and it seems to be now that I’ve read on), the authors might hint at this middle path when introducing the initial distinction between weak and strong sustainability
While there’s no need to elaborate on it, the authors should probably cite Crippen’s (2021) “Africapitalism, Ubuntu and Sustainability” (in Environmental Ethics). Though it’s obviously looking at African philosophy, it applies African traditions to current sustainability issues in Africa, supplying its own empirical data from several case studies, so it’s kind of a precedent to the authors’ manuscript. It also argues for simultaneously anthropocentric and cosmocentric outlooks, so gets close to the authors’ concept of “anthropocosmic.” Finally, it explains hierarchical relationships in which we can have more obligation to family than to non-family, and to humans more than non-humans. Yet it further explains this need not reduce nature or anything in it to a mere instrumentality, lacking in inherent value. This gets close to the authors’ Confucian view. Again, no need to go into a lot of detail, but seems remiss not to cite.
Lastly, the stuff on the Dujiangyan Hydraulic Project is really fascinating, more so because of the longstanding history the authors’ mention. I think more detail is warranted here, both the historical and contemporary sides, as well as a little more exposition on how this connects to Confucian outlooks.
Again, it’s a good paper and I think the changes can be handled fairly easily, but also think that most of them are necessary.
Good quality English, but some minor changes suggested.
Reviewer 2 Report
The article is well-written and the argument is innovative. I only have some minor comments:
1. The author mentions rites are a Confucian method to support sustainability (p. 7). But how can it be? It would be great if the author could give some examples of how rites (especially Confucian rites) can enhance sustainability.
2. The argument of "the unity of heaven and human" is convincing from the perspective of the theorists who endorse strong sustainability. But for those theorists who endorse weak sustainability, they may still ask, "WHY should I extend my care from human beings to all things?" Confucianism merely teaches us how to extend, but why? Why is moral cultivation (p. 5) related to the unity of heaven and human so important? It would be great if the author can explain how Confucians would justify this.
3. Recently there is also an article using the Confucian idea of the unity of heaven and human to address environmental problem. The author might find this book chapter relevant and helpful:
Baldwin Wong, "Pursuing Unity or Creating Disunity? An East-West Complementary Approach to Urben Controversies related to the Right to Environment", in Betty Yung et al. (ed.), Rights and Urban Controversies in Hong Kong: From the Eastern and Western Perspectives (Springer, 2022)
https://link.springer.com/chapter/10.1007/978-981-99-1272-8_10
Round 2
Reviewer 1 Report
The first draft of the paper I read was already good, and I’m please to suggest accepting this one with minor revisions.
But (!!!) this is on the condition that the authors ensure that all or at least most of the suggestions are addressed, since some of them are important and not addressing them would reflect poorly on the authors and the journal.
As a side note, hope this goes into print by November, as I would use it in a class I’m teaching – so a compliment to the authors!
*Suggestions to Authors*
9-10 Suggest changing “is part of the root causes of the human conquest” to something like “has contributed to the human subjugation”.
53-56 – grammatically, I think you should have question marks here.
60- Suggest changing “is one of the root causes leading to the conquest and plundering” to “has arguably contributed to the plundering”
68-69: I imagine the copy editors will take care of it, but suggest moving the parenthetical to abbreviate sentence in: “Crippen has enriched this theory from the perspective of African philosophical traditions. (Crippen 2021)”. That is, you can write: “Crippen (2021) has enriched this theory from the perspective of African philosophical traditions.”
234-241: The passage below NEEDS amending:
Confucianism also believes that nature is a prerequisite for human existence, but that humans can regulate and constrain nature. Xunzi emphasized that humans should control the order of heaven and use it to nourish and transform living creatures (De Bary et al.1999, p.169). In Xunzi's view, human's role concerning heaven is not expressed in a confrontation between heaven and human, but in human's domination of heaven according to the laws of nature itself (De Bary et al.1999, p.174). It is evident 239 that Confucianism advocates that human should live in harmony with nature, respect nature, protect the ecological environment, and use natural resources in a rational manner.
Multiple issues… One is that by the definition of “laws of nature” you gave in your written response , which is more or less laws as physicists see them, everything—including the plundering of nature—occurs within the law of nature. Also, the word “domination” has meanings in English you may not want (generally suggests forceful, unbending or violent control). Finally, a direct supporting quotation from Xunzi would be helpful here.
I would suggest something like changing laws of nature to something like “in harmony with nature. This may still leave a problem, though, because Xunzi, insofar as I understand him, claims that human nature is evil and sees this is why humans need a lot of cultivation. On this account, would acting according to nature mean being evil for humans?
256 – You write: “The term "human" refers to individuals other than oneself…” Why doesn’t the word “human” include oneself?
218 – please clarify what’s meant by “sacrificial activities.” I know China (like many places) practiced human sacrifice at points and in some region (e.g., Sichuan, which is in fact a region you’re dealing with). This is of course very early, but if you’re talking about human sacrifice, then I’d think you want to explain how people transitioned out of that. Even if you’re talking about animal sacrifice, you’d want to explain how this isn’t plundering nature (and it doesn’t have to be plundering since animal sacrifice, in many contexts are eaten, and eating things is hard to escape for living beings). If you just mean something life behaving in a giving (i.e., self-sacrificing way), then I’d suggest changing the wording.
404-466. I’d still really like to hear more about the Dujiangyan Water Hydro-Project and its environmental impact (or how it’s been in harmony with the environment) over time. An added thread you might introduce, is that the Dujiangyan area is constructed with an eye to environmental aesthetics, which can be seen as another form of harmony. One issue you might want to address is how a “regulated and standardized” waterway (456) is in harmony with nature. Heidegger uses just such an example—i.e., the Rhine’s waters being regulated and standardized by dam—as an example of how we subjugate the nature. I think part of Heidegger’s critique might be dispensed with by the general sense in much of Chinese philosophy that humans are always embedded in nature in the first place and that nature does not mean "unaffected or uncultivated" by humans (it’s not that Heidegger denies this, but still the stress, register or tone in Chinese philosophy tends to be different). At the same time (!!), Heidegger will be familiar to a lot of Westerners doing eco-philosophy, so might be worth a few lines. For easy two-page introduction to Heidegger’s views here, I suggest reading pp. 14-16 of “Ecology and Technological Enframement: Cities, Networks, and the COVID-19 Pandemic,” which is posted with original pagination on ResearchGate (which is accessible in China).
Regarding the fact that some gendered language was a consequence of translations, etc., I was aware of that. I am also aware of the particularly complex debates about how 人 should be rendered in English. This is another issue that the journal copy editors will probably address, but, in terms of the alterations made, one gets the sense that the authors’ used a search and replace feature because the grammar doesn’t always fit, e.g., sometimes it should be humans and not human. Also, personally I think a little more word variation would not be confusing to either Chinese readers or English non-experts, so suggest (but leave it up the authors/editors) that variation is added. Other words that could be used are person/people, individual/individuals, in some cases agent/agents, etc.
In any case, at the very least, the grammar should be fixed. To give just one example, “human” should be “humans” in sentence “In contrast, ‘strong sustainability’ emphasizes the long-term sustainable development of human and nature based on maintaining ecological integrity” (97-98).
It needs some moderate work, but think it's also in a state the the copy editor will not have trouble making corrections, i.e., one can tell what the authors are trying to say almost all the time and the sentences are mostly structurally fine, so it's mainly just small things like conjugation changes that are needed.
